# Collision Avoidance Metric for 3D Camera Evaluation

Vage Taamazyan          Alberto Dall'olio          Agastya Kalra

Intrinsic Innovation LLC

## Abstract

*3D cameras have emerged as a critical source of information for applications in robotics and autonomous driving. These cameras provide robots with the ability to capture and utilize point clouds, enabling them to navigate their surroundings and avoid collisions with other objects. However, current standard camera evaluation metrics often fail to consider the specific application context. These metrics typically focus on measures like Chamfer distance (CD) or Earth Mover's Distance (EMD), which may not directly translate to performance in real-world scenarios. To address this limitation, we propose a novel metric for point cloud evaluation, specifically designed to assess the suitability of 3D cameras for the critical task of collision avoidance. This metric incorporates application-specific considerations and provides a more accurate measure of a camera's effectiveness in ensuring safe robot navigation.*

## 1. Introduction

Almost every existing robotics or autonomous driving system relies on a 3D scene representation for functions such as object detection, path planning, and overall scene understanding [11, 14, 18, 23, 24, 30]. This representation typically takes the form of a point cloud, acquired using sensors like LiDAR, time-of-flight, structured light, or stereo cameras. The 3D scene representation is crucial for the robot to navigate safely and perform its tasks without colliding with objects [8]. Therefore, the quality of the point cloud is essential, and evaluating it becomes paramount.

However, current point cloud evaluation methods often assess the point cloud itself, neglecting its downstream application. Metrics like Chamfer Distance (CD) [5, 7], Hausdorff Distance [4, 6], Earth Mover's Distance (EMD) [21], Completeness and Accuracy [10] are commonly employed, but they solely evaluate point cloud properties and don't directly translate to application performance. Furthermore, comparing sensors solely based on these metrics can be misleading, as one sensor might excel in one metric while another dominates in another metric. We argue that existing

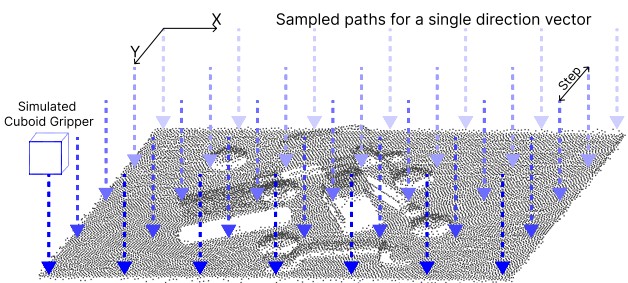

Figure 1. Visualization of the Collision Avoidance Metric evaluation process. A set of grippers moving towards the point cloud is simulated with the same initial direction vector sampled on the XY plane with step S until collision is detected.

metrics fail to directly capture the core purpose of the point cloud: enabling safe and successful operation.

To address this gap, we propose the Collision Avoidance metric. This new metric offers several key advantages:

1. **Addresses Downstream Performance:** This metric directly assesses the suitability of a point cloud from a specific 3D camera for a particular robotics or autonomous driving application. It focuses on downstream performance, meaning it goes beyond the point cloud itself and evaluates its effectiveness in real-world tasks. See Sec. 3 for more details.

2. **Supports Threshold Identification:** The metric can be used to identify various tolerances for collision detection that can then be programmed into the robot. This enables the robot to adjust its sensitivity based on the specific environment and sensor capabilities, leading to better adaptation and performance. See Sec. 4.5 for more details.

3. **Interpretable:** Points of missed or incorrectly detected collisions are clearly highlighted, aiding in understanding what contributes to the error and to what extent. See Sec. 4.3 for more details.

Our intuition behind the metric is straightforward: ide-

ally, we would like to evaluate a point cloud's effectiveness by testing a robot in various real-world scenarios. This would involve capturing a point cloud of an arbitrary scene, placing a real robot in a large number of random locations, and checking for collisions. However, this approach is impractical and potentially damaging due to its cost, complexity, and potential for causing damage during collisions.

As a more feasible alternative, a simulation-based approach could be used. If both captured and ground truth point clouds are available, we can simulate various robot movements and compare the predicted and actual collisions. While this method is less ideal as it relies on ground truth accuracy, it offers a more realistic evaluation compared to solely evaluating point cloud properties. However, implementing such simulations can become complex.

Therefore, we introduce an additional simplification: we assume a rectangular cuboid shaped gripper of size $L \times M \times N$ with linear movement. We move the gripper along a direction $\vec{x}$ until a collision is detected with the predicted point cloud and the ground truth point cloud. This allows us to compare the distance to collision between both scenarios. We repeat this process for various directions and convert these comparisons into False Positive and False Negative Collision rates for predicted collisions. One of the benefits of this approach is its ability to inform robot tolerance requirements. For instance, we can increase the cube size until false negative collisions are eliminated, providing insights into the necessary safety margins for the robot.

We demonstrate, using captured scenes with various 3D sensors, that the proposed Collision Avoidance metric effectively ranks 3D sensors for specific applications based on their suitability for those tasks. In summary, our key contributions are:

1. We propose a novel Collision Avoidance metric for evaluating point clouds in robotic and autonomous driving applications.
2. We compare our metric to existing methods and demonstrate, using a variety of captured scenes with diverse materials and lighting conditions, how effectively the metric ranks different 3D sensors based on their performance.

While evaluating a sensor requires capturing raw 3D data using that sensor, we do not propose a specific dataset. We believe the ease of replicating the method with any point cloud makes it widely applicable and readily adoptable.

## 2. Related Work

The evaluation of point clouds predominantly employs metrics such as the Chamfer Distance [5, 7] and its variations, including the Density-aware Chamfer Distance [29], the Hausdorff Distance [4, 6], and the Earth Mover's Distance (EMD) [21]. These metrics have been widely used in a significant body of research related to point cloud evaluation [2, 7, 16], with their differentiability making them suitable for direct use as loss functions during the training process [7, 20].

In robotics, especially in applications involving LiDAR technology, metrics based on the Root Mean Squared Error (RMSE), such as the point-to-point error, are frequently utilized for evaluating point clouds [13, 15, 17, 25].

Other metrics, albeit less commonly employed, include the Structural Similarity Index Metric (SSIM) [3], which assesses point clouds based on geometric or color features, and is less often used for direct comparisons of point clouds captured by 3D sensors. Another example is DPDist, which compares point clouds by measuring the distance between the surfaces from which they were sampled [26]. The Sliced Wasserstein distance [19] is mentioned as well, noted for properties similar to those of the EMD.

The following sections provide a more detailed examination of the commonly utilized metrics.

### 2.1. Chamfer Distance

The Chamfer Distance (sometimes called the Chamfer Measure) has several formulations, but they all share a core principle. For each point in a point cloud $P_1$ of size N, the distance to the closest point in another point cloud $P_2$ of size M is found. These distances are then summed. This process is repeated for each point in $P_2$ with respect to the points in $P_1$.

Two common distance measures are employed: the $L_1$ norm and the $L_2$ norm (Euclidean distance) [7]. Additionally, while sometimes the sum is divided by the number of points in the point cloud [28], this normalization step does not affect the fundamental concept of the approach.

$$CD(P_1, P_2) = \frac{1}{N} \sum_{x \in P_1} \min_{y \in P_2} \|x-y\|_2^2 + \frac{1}{M} \sum_{y \in P_2} \min_{x \in P_1} \|x-y\|_2^2$$

(1)

### 2.2. Hausdorff Distance

The Hausdorff Distance between two point clouds, denoted as $P_1$ and $P_2$, is typically defined as the greatest of all the distances from a point in one set to the closest point in the other set. Specifically, if we define the minimum distance from a point $x$ in $P_2$ to the set $P_1$ as:

$$MD(x, P_2) = \min_{y \in P_2} |x - y|_2,$$

(2)

then the one-sided Hausdorff Distance from $P_1$ to $P_2$ is given by:

$$HD_{os}(P_1, P_2) = \max_{x \in P_1} MD(x, P_2),$$

(3)

where $HD_{os}(P_1, P_2)$ denotes the one-sided Hausdorff Distance from $P_1$ to $P_2$.

The full Hausdorff Distance between $P_1$ and $P_2$ is then defined as the maximum of the two one-sided Hausdorff Distances [4] or their sum [28]. This can be represented as:

$$HD(P_1, P_2) = \max_{x \in P_1} MD(x, P_2) + \max_{y \in P_2} MD(y, P_1) \quad (4)$$

This measure effectively captures the notion of the greatest distance between two point clouds.

## 2.3. Earth Mover's Distance

Assume that point clouds $P_1$ and $P_2$ are of equal size. The Earth Mover's Distance (EMD) between these point clouds can be defined as:

$$EMD(P_1, P_2) = \min_{\phi:P_1 \to P_2} \sum_{x \in P_1} |x - \phi(x)|_2, \quad (5)$$

where $\phi : P_1 \to P_2$ represents a bijection between points in $P_1$ and points in $P_2$ [7].

In more general terms, when the point clouds $P_1$ and $P_2$ do not necessarily have the same size, the correspondence between points in these clouds can be established through a mapping matrix, $\Pi(P_1, P_2)$, an $M \times N$ matrix. In this matrix, the sum of the columns and rows equals one, and each element, $\Pi_{i,j}$, ranging from 0 to 1, specifies the degree to which points $x_i$ from $P_1$ and $y_j$ from $P_2$ correspond to each other [28].

## 2.4. Accuracy, Completeness and F-score

Evaluating point clouds in benchmarks for Multiview Stereo often involves metrics such as accuracy and completeness, commonly referred to as precision and recall, along with their respective F-scores [10, 22, 27]. Among the metrics described, this set uniquely incorporates an explicit distinction between Ground Truth and Reconstructed point clouds. Following the definition by Knapitsch et al. [10], these metrics are defined as follows:

Let $P_{GT}$ represent the ground truth point cloud containing $M$ points, and $P_Q$ denote the captured or reconstructed point cloud with $N$ points, which is subject to evaluation. For any point $x \in P_Q$, the distance to $P_{GT}$ is defined as:

$$D_{x \to P_{GT}} = \min_{y \in P_{GT}} \|x - y\| \quad (6)$$

Subsequently, the precision (or accuracy) metric for a given threshold $d$ is defined as:

$$P(d) = \frac{1}{N} \sum_{x \in P_Q} [D_{x \to P_{GT}} < d] \quad (7)$$

where $[\cdot]$ represents the Iverson bracket, situating $P(d)$ within the range of 0 to 1. Conversely, the recall (or completeness) metric is defined by reversing the roles of $P_{GT}$

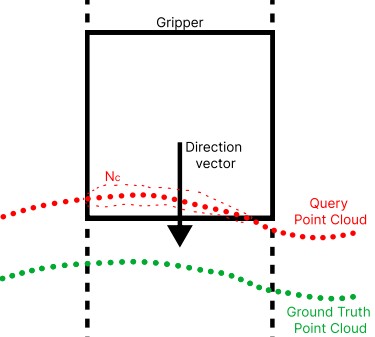

Figure 2. The process of detecting a False Positive Collision on a gripper's path. A collision is initially detected if the number of points inside the gripper $N_c$ is more than a threshold number of points, denoted by $N_{GT}$ for the Ground Truth point cloud and $N_Q$ for the Query point cloud. A collision is considered false positive if: 1. The Query point cloud collides with the gripper before the GT point cloud. 2. The distance between the two collision points is greater than the Z tolerance.

and $P_Q$. For any point $y \in P_{GT}$, the distance to $P_Q$ is defined as:

$$D_{y \to P_Q} = \min_{x \in P_Q} \|x - y\| \quad (8)$$

Thus, the recall for the given threshold $d$ is:

$$R(d) = \frac{1}{M} \sum_{y \in P_{GT}} [D_{y \to P_Q} < d] \quad (9)$$

Typically, precision and recall are combined to compute the F-score via the harmonic mean:

$$F(d) = \frac{2P(d)R(d)}{P(d) + R(d)} \quad (10)$$

## 3. Collision Avoidance Metric Description

This section describes the Collision Avoidance metric for a captured point cloud, also referred to as the "query" point cloud, denoted as $PC_Q$. The metric aims to compare the robot's detected collisions with objects in the scene to what would occur in the real world, using the collisions with ground truth (GT) point cloud as a proxy for real-world collisions. This metric focuses on ensuring the safe and successful operation of the robot in any given scene, regardless of material types or lighting conditions.

The metric identifies two types of errors:

1. **False Positive Collisions (FPC):** Collisions predicted by the query point cloud that wouldn't occur in reality (also called "ghost collisions").
2. **False Negative Collisions (FNC):** Collisions missed by the query point cloud that would occur in reality ("missed collisions").

These errors are aggregated into FPC rate $R_{FPC}$ and FNC rate $R_{FNC}$, as well as Collision F-score $FC$. The metrics are described in detail in Sec. 3.2.

## 3.1. Inputs

The metric requires the following data:

1. $PC_{GT}$: Ground truth point cloud of the scene.
2. $PC_Q$: Query point cloud of the scene captured by a sensor or group of sensors.
3. $T_Z$: Gripper tolerances in Z direction.
4. $L \times M \times N$: Gripper size (length, width, height).
5. $G_{step}$: Gripper step in XY plane. This is also a proxy to XY tolerance, see details in Sec. 3.2.
6. $N_{GT}$ and $N_Q$: Outlier thresholds (number of points exceeding the threshold) for the ground truth and query point clouds, respectively (consistent across all scenes for each sensor).
7. $Ds$: Set of directions for gripper movement. Each direction is a 3D vector.

## 3.2. Algorithm

The evaluation process follows these steps:

1. **Load point cloud:** Load the relevant point cloud (either $PC_{GT}$ or $PC_Q$) and its corresponding outlier threshold.
2. **Iterate over directions:** Iterate through gripper movement directions $Ds$. Each direction is a 3D vector, defining the direction along which the gripper movement is simulated (see Fig. 1 and Fig. 2).
3. **Sample initial positions:** For each direction, sample a set of initial gripper positions in the plane perpendicular to the direction vector, using the $G_{step}$ parameter. This creates a set of final paths for the gripper descent.
4. **Simulate gripper descent:** For each path, simulate the gripper's descent and find the moment of collision, defined as the first instance where the number of points inside the gripper is greater than the threshold value ($N_{GT}$ or $N_Q$).
5. **Label paths:** Repeat steps 1-4 for both $PC_{GT}$ and $PC_Q$. Each path of the ground truth is compared with corresponding path of the query and its 4 neighbor paths is XY plane, and the best output is selected. This step defines the $T_{XY}$ as the $G_{step}$ defined above. Then every path is labeled as follows:
   (a) **Aligned**: $PC_Q$ point cloud is on par with $PC_{GT}$ point cloud for collision avoidance for that path.
   (b) **FPC**: $PC_Q$ predicts a collision that wouldn't occur based on $PC_{GT}$ within the $T_Z$ from the collision point (ghost collision).
   (c) **FNC**: $PC_GT$ predicts a collision that wouldn't occur based on $PC_Q$ within the $T_Z$ from the collision point (missed collision).
6. **Calculate rates:** Calculate the FPC and FNC rates as the percentage of number of FPC and FNC collisions relative to the total number of paths across all directions.

$$R_{FNC} = \frac{N_{FNC}}{N_{Total}} \qquad (11)$$

$$R_{FPC} = \frac{N_{FPC}}{N_{Total}} \qquad (12)$$

where $N_{FNC}$ is the number of paths labeled as False Negative Collisions, $N_{FPC}$ is the number of paths labeled as False Positive Collisions, $N_{Total}$ is the total number of paths. The Collision F-score is then defined as:

$$FC = 1 - \frac{2(1 - R_{FNC})(1 - R_{FPC})}{2 - R_{FNC} - R_{FPC}} \qquad (13)$$

7. **Tolerance analysis:** (Optional) Repeat steps 1-6 for different $T_Z$ tolerances to determine the optimal tolerance for a specific application.

## 3.3. Discussion

This section compares the proposed metric to the existing metrics discussed in Sec. 2, highlighting its key conceptual differences:

1. **Target Application:** The proposed metric is specifically tailored for industrial point cloud evaluations: it penalizes omissions that could impact real-world robotic applications. For example, if a thin structure like a screwdriver is omitted from the point cloud, this wouldn't significantly change any of the metrics described in Sec. 2, but would be penalized more strongly by the Collision Avoidance metric. We believe this is important because such an omitted screwdriver could cause a collision with the robot and break the gripper. See Fig. 3 for an example of how the Collision Avoidance metric reacts to small objects missed in the point clouds (compared to other metrics).
2. **Density Tolerance:** Unlike existing metrics, this metric doesn't penalize a less dense query point cloud ($P_Q$) compared to the ground truth ($P_{GT}$) as long as the captured density fulfills the specific gripper/robot endeffector specifications. This focuses on functionality rather than absolute density.
3. **Small Hole Tolerance:** Similar to point 1, the metric doesn't penalize small holes in the query point cloud, as long as they are smaller than the robot end-effector. This prevents penalizing the metric for irrelevant details smaller than the gripper/robot's operational size.
4. **Directional Tolerance:** The metric allows independent tolerances for XY and Z dimensions based on a given direction vector. This acknowledges the inherent differences in resolution and importance depending on the application and robot movement.
5. **Combined Completeness and Accuracy:** This metric effectively merges the concepts of completeness and accuracy from Sec. 2.4. High FNC often indicate incomplete data, while both FNC and FPC can arise from inaccurate data depending on the type of inaccuracy.

This combined approach provides a more comprehensive evaluation.

6. **Directionality:** Unlike completeness and accuracy metrics from Sec. 2.4, the Collision Avoidance metric explicitly considers directions, further enhancing its relevance for downstream applications. For example, it may be more important to capture a depth map and ensure safe robot movements along a given direction, which would be less sensitive to certain types of holes. We show the metric dependence on directions in Sec. 4.5.

# 4. Evaluation

This section demonstrates how our proposed metric can be used to evaluate a structured light sensor (Photoneo XL [1]) against active and passive stereo cameras. When choosing 3D cameras for robotics, there are often trade-offs to consider. Structured light sensors typically offer higher accuracy but lower completeness due to occlusions caused by large baselines and challenging surfaces. Passive stereo cameras can handle these issues better but may have lower accuracy. We aim to demonstrate how the Collision Avoidance metric can be used to compare these cameras effectively, considering the specific requirements of the downstream robotic application.

We utilize two stereo depth reconstruction algorithms, CREStereo [12] and SGM [9], to generate point clouds from the stereo camera data. Details on the test scenes capture are provided in Sec. 4.1. Our goal is to rank the 3D cameras based on a set of defined tolerances, as discussed in Sec. 4.2. We also separately test small ($\sim$10 cm) and large ($\sim$100 cm) baselines for stereo cameras, as shown in Sec. 4.4. Additionally, in Sec. 4.5, we further explore how the chosen tolerances and other input parameters (introduced in Sec. 3.1) can influence the ranking.

## 4.1. Test Scenes

We captured a set of scenes using a stereo camera in both active and passive modes, alongside a Photoneo XL structured light 3D sensor. The active stereo camera projects an infrared (IR) dot pattern onto the scene, providing additional texture for the stereo matching algorithm. This study utilizes CREStereo and SGM as the stereo matching algorithms. The scenes encompass objects with diverse sizes, shapes, and material properties, captured under both room light and sunlight conditions.

To obtain a ground truth ($P_{GT}$) point cloud of a scene, we implemented the following procedure:
1. Mount the scene onto a robot arm.
2. Compute hand-eye calibration between the Photoneo and the robot arm.
3. Apply a coat of white matte paint, specifically formulated for 3D object scanning, to the scene.

4. Utilize the robot arm to rotate the scene and capture a point cloud from each position using Photoneo.
5. Merge the captured point clouds into a single, complete, and accurate point cloud knowing the robot intermediate joint configurations.

This outlined procedure guarantees an accurate and complete point cloud representation of the scene, regardless of the material properties or lighting conditions. To capture a set of test scenes, we allowed the spray paint to dry completely before replicating the robot arm positions under both room light and sunlight. Since the robot is repeating the pose with sub-millimeter accuracy, it doesn't create a significant additional error. During this process, we captured 3D data with each sensor in the scene. To ensure well-aligned captured point clouds with the rotated GT point cloud, we positioned the stereo cameras and Photoneo XL as close as possible to each other and performed pre-calibration among them.

The evaluation produces 5 different point clouds:
- Photoneo point cloud obtained directly from the 3D sensor;
- Point cloud obtained by the SGM algorithm using the passive stereo setup with RGB camera stereo pair;
- Point cloud obtained by the SGM algorithm using the active stereo setup with IR camera stereo pair;
- Point cloud obtained by the CREStereo algorithm using the passive stereo setup with RGB camera stereo pair;
- Point cloud obtained by the CREStereo algorithm using the active stereo setup with IR camera stereo pair;

The metrics outlined in Sec. 2, alongside the novel Collision Avoidance metric, are computed based on the Ground Truth point cloud generated as described above. The comparison between the Collision Avoidance metric and existing ones is presented in Tab. 2. Furthermore, Tab. 1 illustrates the performance of Collision Avoidance metric under various lighting conditions, demonstrating its effectiveness in identifying challenges faced by 3D structured light scanners in intense lighting scenarios.

## 4.2. Results

We set the following values to the parameters from Sec. 3.1: $T_Z = 10mm$, $L \times M \times N = 10mm \times 10mm \times 10mm$, $G_{step} = 5mm$, $N_{GT} = 15$, $N_Q = 5$, $Ds = [0, 0, 1]$. Tab. 1 shows the aggregated $R_{FPC}$, $R_{FNC}$ and $FC$ for the tested sensors under both room light and sunlight conditions. We evaluate the performance considering only a single direction, from the top down (Z-axis). Sec. 4.5 analyzes how the metrics change when additional directions are included. For this study, we focus on evaluating the point cloud's suitability for vertical robot movements. Additionally, we ensure that the evaluated cameras are positioned close together, with their direction vectors aligned with the Z-axis. Fig. 4 demonstrates how all the path collision points including the

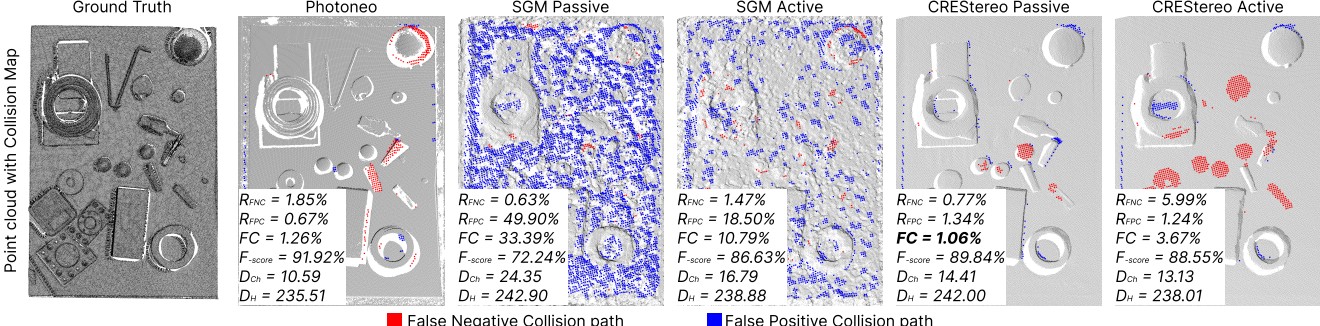

Figure 3. A sample scene from the evaluation dataset. Collision maps are aligned with each point cloud. Red dots indicate FNC paths: collisions missed by the captured point cloud, while blue dots represent FPC paths: "ghost" collisions detected by the captured point cloud. Note that the CREStereo Active has lower Chamfer and Hausdorff distances than CREStereo Passive, but it also has significantly higher $R_{FNC}$ and $FC$. These metrics alone would not reveal the increased risk of collision for the robot gripper in this scene. Similarly, Photoneo achieves a better F-score and lower Chamfer and Hausdorff distances compared to CREStereo Passive, but performs slightly worse in $FC$ again, primarily due to its higher $R_{FNC}$. For optimal viewing, zoom in on a digital copy of the paper.

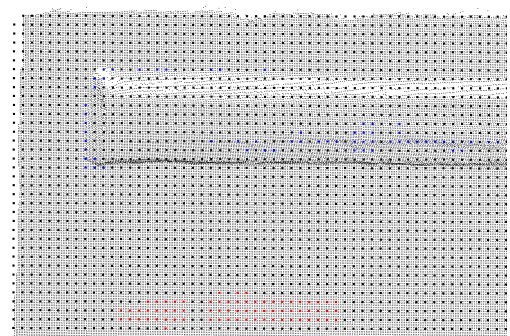

Figure 4. A crop of a point cloud with simulated paths. Black dots are the paths with Aligned label from Sec. 3.2, blue dots - the FPC paths, red dots - the FNC paths.

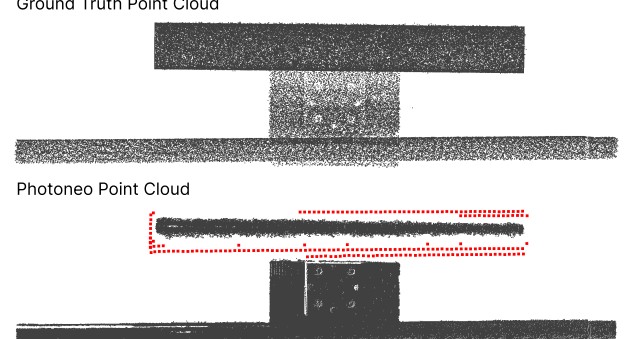

Figure 5. The red dots represent FNC points, indicating missed collisions during a specific gripper movement direction. Capturing metallic surfaces, like this pipe, can be challenging for structured light 3D scanners.

FP and FN Collisions are spanned over the point cloud for a given gripper direction.

Tab. 2 aggregates the metrics for all the point clouds across all the scenes. Several important conclusions can be drawn from the table:

1. **Chamfer and Hausdorff metrics are not well-suited for evaluating how 3D sensors compare in terms of collision avoidance.** SGM Active has a lower Chamfer distance than Photoneo and CRES Passive, however the FC metric shows that SGM Active would basically be unusable for any robotic application with 41.43% $R_{FPC}$. Similarly, Hausdorff distance is very similar for CRES Active and Photoneo, while the $R_{FNC}$ is much higher for Photoneo, making its use in industrial applications for Collision Avoidance much riskier.

2. **Completeness, Accuracy, and F-Score provide results more aligned with the Collision Avoidance metric, as expected due to their conceptual similarity.** However, the Collision Avoidance metric provides a clearer inter-

pretation and more actionable insights due to its focus on real-world applications. Additionally, it exhibits greater sensitivity to even minor yet critical discrepancies for the robot, as demonstrated in Fig. 3.

3. **We argue that most robotic applications prioritize a low $R_{FNC}$ while maintaining a reasonable $R_{FPC}$.** A low $R_{FNC}$ ensures the robot avoids collisions, while a moderate $R_{FPC}$ maintains operational efficiency. Therefore, the Collision Avoidance metric provides the most valuable data for ranking sensors and algorithms for specific downstream robotic applications.

### 4.3. Interpretability

The metric directly identifies missed and ghost collision points, allowing their visualization on the query point cloud (see Figs. 3 to 5 for an example). Each red dot represents a FNC, indicating a collision missed by the query point cloud

| 3D sensor | $R_{FPC}$ **room light [%]** ↓ | $R_{FNC}$ **room light [%]** ↓ | $FC$ **room light [%]** ↓ | $R_{FPC}$ **sunlight [%]** ↓ | $R_{FNC}$ **sunlight [%]** ↓ | $FC$ **sunlight [%]** ↓ |
|---|---|---|---|---|---|---|
| Photoneo XL | 0.58 | 3.50 | **2.06** | 0.10 | 18.30 | 10.11 |
| SGM Passive | 51.82 | 8.00 | 36.76 | 53.99 | 10.22 | 39.16 |
| SGM Active | 36.29 | 1.1 | 22.51 | 46.62 | 1.21 | 30.69 |
| CRES Passive | 2.71 | 9.74 | 6.36 | 3.08 | 12.54 | 8.06 |
| CRES Active | 1.97 | 3.42 | 2.7 | 1.29 | 1.03 | **1.16** |

Table 1. Results of collision avoidance metric under different conditions. This table shows how the proposed metrics can rank different algorithms based on different light conditions. While Photoneo XL performs best in normal light, Active CRES exhibits greater stability across various lighting conditions, particularly under direct sunlight. ↓ indicates that lower is better.

| 3D sensor | $R_{FPC}$ **[%]** ↓ | $R_{FNC}$ **[%]** ↓ | $FC$ **[%]** ↓ | **Chamfer [mm]** ↓ | **Hausdorff [mm]** ↓ | **Completeness [%]** ↑ | **Accuracy [%]** ↑ | **F-Score [%]** ↑ |
|---|---|---|---|---|---|---|---|---|
| Photoneo XL | 0.33 | 10.89 | 5.90 | 16.38 | 92.82 | 83.01 | 98.45 | 89.60 |
| SGM Passive | 52.91 | 9.11 | 37.96 | 21.58 | 132.84 | 86.23 | 66.44 | 74.22 |
| SGM Active | 41.43 | 1.15 | 26.45 | 14.05 | 107.83 | 92.99 | 79.77 | 85.72 |
| CRES Passive | 2.91 | 11.15 | 7.21 | 16.21 | 98.58 | 84.21 | 93.32 | 88.31 |
| CRES Active | 1.63 | 2.23 | 1.93 | 12.33 | 90.99 | 90.67 | 96.67 | 93.50 |

Table 2. Metrics comparison. This table shows the behaviour of the proposed Collision Avoidance metric compared to the already existing metrics. ↓ indicates that lower is better, ↑ indicates that higher is better.

despite occurring in reality. Conversely, each blue dot represents a FPC, signifying a collision predicted by the system but absent in the real world.

Fig. 5 demonstrates how Photoneo struggles to capture some shiny pipe sections in sunlight conditions, leading to a higher number of FNCs. These False Negative Collisions occur on the borders of the pipe, in regions not properly reconstructed by the sensor.

## 4.4. Analysis of Different Stereo Baselines

This subsection delves deeper into understanding Collision Avoidance metric by examining the effects of varying stereo baselines on Passive Stereo SGM performance. Generally, increasing the camera baseline improves the accuracy of the generated point cloud. However, this comes at the cost of reduced completeness due to potential occlusions caused by viewpoint differences. Conversely, a smaller baseline results in a more complete point cloud, but with increased noise.

This trade-off between completeness and accuracy is reflected in the Collision Avoidance metric presented in Tab. 3. As the baseline increases, $R_{FPC}$ decreases by approximately 40%, indicating reduced noise. However, $R_{FNC}$ roughly doubles, highlighting the completeness issues in the generated point cloud. Interestingly, the Chamfer and Hausdorff distances show less sensitivity to baseline changes, while the Accuracy and Completeness metrics exhibit a more noticeable difference (approximately 10%) with the larger baseline approach even though the F-Score results remains the same.

## 4.5. Input Parameters Ablation

The proposed metric can be sensitive to the input parameters. This subsection analyzes how 3D sensor rankings would change depending on these parameters. We focus on two parameters most relevant for real-world robotic applications:

1. **Gripper Tolerance Ablation** ($T_Z$)**:** This ablation investigates how Collision Avoidance metric changes according to the required tolerance. Using such analysis, user can evaluate a point cloud's suitability for collision avoidance in a specific application. For example, if an application requires a certain $R_{FNC}$, the user can define a gripper tolerance that guarantees the expected performance.
2. **Gripper Direction Ablation** ($Ds$)**:** The results in Sec. 4.2 consider only one possible gripper descent direction. This ablation shows how the metrics change if multiple directions are used.

We conducted the initial ablation by computing the metrics with eight different tolerance values for $T_Z$ ranging from 2.5 to 20.0 millimeters. The results of this ablation on $T_Z$ are shown in Fig. 6 and Fig. 7. As the tolerance increases, the metrics become more permissive. For example, the $R_{FPC}$ for Active SGM approaches and ends up being better than Photoneo by increasing the threshold. This observation indicates that as the tolerance becomes more permissive, the noise level of the point cloud becomes less critical. Fig. 6 illustrates that this parameter has smaller impact on $R_{FNC}$ in the test scenes, suggesting that adjusting gripper tolerance may not necessarily improve the outcome if the query point cloud fails to capture certain areas of the scene. However, it is important to note that this analysis is scene-dependent and the order may change for other scenes and 3D cameras.

To explore the impact of gripper directions, we computed the collision metric using 1, 4, and 7 distinct directions for the simulated direction vectors on Photoneo point clouds. This analysis expands the gripper's exploration beyond simply facing perpendicular to the point cloud plane

| 3D sensor | $R_{FPC}$ [%] ↓ | $R_{FNC}$ [%] ↓ | $FC$ [%] ↓ | Chamfer [mm] ↓ | Hausdorff [mm] ↓ | Completeness [%] ↑ | Accuracy [%] ↑ | F-Score [%] ↑ |
|---|---|---|---|---|---|---|---|---|
| SGM 10 cm Baseline | 52.91 | 9.11 | 37.96 | 21.58 | 132.84 | 86.23 | 66.44 | 74.22 |
| SGM 100 cm Baseline | 29.13 | 15.07 | 22.74 | 24.02 | 127.08 | 76.27 | 76.83 | 74.90 |

Table 3. Comparison of metrics under varying stereo baselines for the same algorithm. The tables present results of both proposed and existing metrics applied to the same stereo algorithm with two distinct baselines. These results highlight the effectiveness of the proposed Collision Avoidance metric in evaluating denser, noisier point clouds with smaller baselines, and sparser, more accurate point clouds with larger baselines. ↓ indicates that lower is better. ↑ indicates that higher is better.

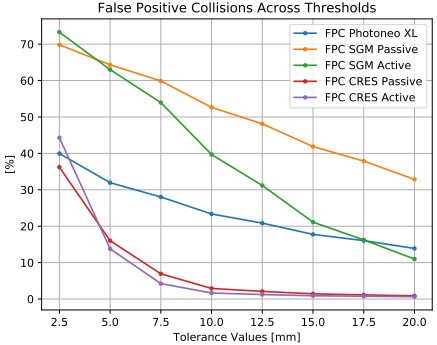

Figure 6. $R_{FPC}$ comparison across increasing values of Z tolerance ($T_Z$). With a higher tolerance, the Collision Avoidance metric becomes more permissive and the 3D sensors ranking changes, positioning Active SGM to be better than Photoneo for a $T_Z = 20mm$.

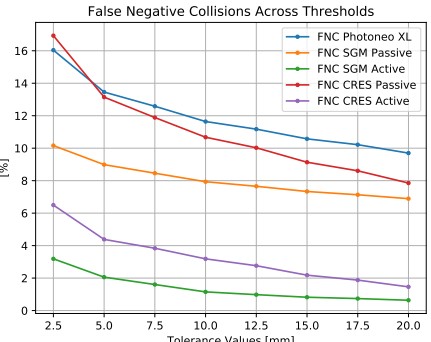

Figure 7. $R_{FNC}$ comparison across increasing values of Z tolerance ($T_Z$). $R_{FNC}$ is less influenced by different gripper tolerances values in the set of scenes used for evaluation.

(assuming a top-down view). For the 4-direction case, we introduce 3 additional directions tilted 30 degrees from perpendicular. Similarly, for 7 directions, we introduce 3 additional directions tilted 45 degrees.

As discussed in Sec. 4.2, increasing the number of gripper directions has minimal impact on the metrics with a small stereo baseline and when the descent direction aligns with the cameras' Z axis. However, this effect might not hold true for larger baselines, as seen with the Photoneo

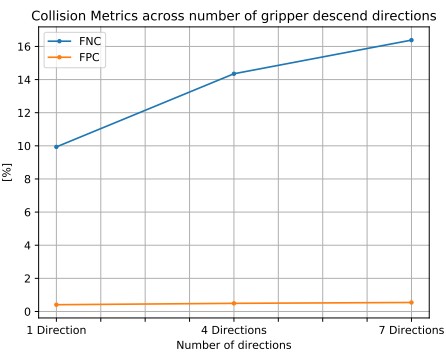

Figure 8. Variation in Collision Avoidance metric with an increasing number of gripper descent directions for Photoneo XL point clouds. The increase in $R_{FNC}$ can be explained by the limitations of capturing top-down scenes with Photoneo, which might lead to occluded and unreconstructed areas.

XL camera. Fig. 8 demonstrates a significant increase (over 50%) in the $R_{FNC}$ for Photoneo XL. This can be attributed to potential occlusions in the top-down view of the point cloud, leading to missed regions. By incorporating additional directions, we detect these occluded areas.

## 5. Conclusions

This work introduces the Collision Avoidance metric, a novel approach to point cloud evaluation that prioritizes both downstream performance and interpretability. We demonstrate its intuitiveness and effectiveness in evaluating point clouds from active and passive 3D sensors, particularly for robotic applications. Comparisons to other point cloud evaluation metrics show that the Collision Avoidance metric can lead to better interpretability and different 3D camera rankings that may be more relevant to downstream applications focused on collision avoidance. Extensive ablation studies validate the impact of the metric's parameters. We believe the Collision Avoidance metric will empower researchers and engineers to more effectively assess the suitability of point clouds for tasks requiring robust collision avoidance, as well as help them find optimal parameters for their systems.

**Acknowledgement.** We would like to thank Kartik Venkataraman for his general support and helpful discussions.

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
