# OpenReview forum: "Collision Avoidance Metric for 3D Camera Evaluation"
_thecvf.com/CVPR/2024/Workshop/VLADR — VLADR 2024 Oral_

### Official Review · Reviewer_7Reb · 2024-04-17

**Rating:** 7
**Confidence:** 5

**Review:**

Overall Evaluation:

The paper proposes a novel and well-designed metric for evaluating 3D cameras specifically for their suitability in robotic and autonomous driving applications focused on collision avoidance. The metric goes beyond just evaluating the point cloud properties and directly assesses the camera's effectiveness in enabling safe robot navigation. This is an important contribution, as existing evaluation metrics often fail to capture the actual downstream performance required for real-world applications.

Strengths:

- The Collision Avoidance metric provides a more application-oriented evaluation that focuses on the core purpose of 3D cameras - enabling safe robot operation. This is a significant improvement over generic point cloud metrics.

- The metric can be used to identify optimal tolerances for collision detection, allowing robots to better adapt to the sensor capabilities and environmental conditions.

- The interpretability of the metric, with clear identification of false positive and false negative collision points, is very useful for understanding and improving sensor performance.

- The thorough evaluation against various 3D sensors and stereo camera baselines demonstrates the effectiveness and versatility of the proposed metric.

Weaknesses:

- While the paper discusses the intuition behind the metric, more formal mathematical formulation and justification could strengthen the presentation.

- The paper could benefit from a more detailed analysis of the impact of the input parameters (e.g., gripper size, tolerance values) on the final rankings, beyond the ablation study provided.

- Comparison to other application-specific metrics, if any exist in the literature, could provide additional context and insights.

Recommendation:

Overall, this is a well-executed and impactful work that addresses an important gap in 3D camera evaluation. I recommend accepting this paper for publication, as it provides a valuable new tool for the robotics and autonomous driving communities.

---

### Decision · Program_Chairs · 2024-04-22

Accept (Oral)